# Prevalence and factors associated with long COVID and mental health status among recovered COVID-19 patients in southern Thailand

Doan Hoang Phu[1], Sarunya Maneerattanasak[2], Shamarina Shohaimi[3], Le Thanh Thao Trang[4,5], Truong Thanh Nam[5,6], Muminah Kuning[7], Aunchalee Like[7], Hameedah Torpor[7], Charuai Suwanbamrung[5,8]*

1 Doctoral Program in Health Sciences, College of Graduate Studies, Walailak University, Nakhon Si Thammarat, Thailand, 2 Department of Immunology, Faculty of Medicine Siriraj Hospital, Mahidol University, Bangkok, Thailand, 3 Department of Biology, Faculty of Science, Universiti Putra Malaysia, Malaysia, Malaysia, 4 Faculty of Basic Science and Public Health, Dong Thap Medical College, Cao Lanh City, Dong Thap, Vietnam, 5 M.P.H. and Ph.D. Program in Public Health Research, School of Public Health, Walailak University, Nakhon Si Thammarat, Thailand, 6 Faculty of Public Health, Can Tho University of Medicine and Pharmacy, Can Tho City, Vietnam, 7 Community Public Health Program, School of Public Health, Walailak University, Nakhon Si Thammarat, Thailand, 8 Excellent Center for Dengue and Community Public Health (EC for DACH), Walailak University, Nakhon Si Thammarat, Thailand

* yincharuai@gmail.com

**Data Availability Statement:** All relevant data are within the manuscript and its Supporting Information files.

## Abstract

Mental health disorders have become a growing public health concern among individuals recovering from COVID-19. Long COVID, a condition where symptoms persist for an extended period, can predict psychological problems among COVID-19 patients. This study aimed to investigate the prevalence of long COVID and mental health status among Thai adults who had recovered from COVID-19, identify the association between the mental health status and long COVID symptoms, and investigate the risk factors associated with the correlation between long COVID and mental health outcomes. A cross-sectional study was conducted among 939 randomly selected participants in Nakhon Si Thammarat province, southern Thailand. The Depression, Anxiety, and Stress Scale-21 was used to investigate mental health symptoms, and a checklist comprised of thirteen common symptoms was used to identify the long COVID among participants. Logistic regression models were used to investigate the risk factors associated with mental health status and long COVID symptoms among participants. Among the 939 participants, 104 (11.1%) had depression, 179 (19.1%) had anxiety, and 42 (4.8%) were stressed. A total of 745 participants (79.3%) reported experiencing at least one symptom of long COVID, with fatigue (72.9%, SE±0.02), cough (66.0%, SE±0.02), and muscle pain (54.1%, SE±0.02) being the most frequently reported symptoms. All long COVID symptoms were significantly associated with mental health status. Shortness of breath, fatigue, and chest tightness were the highest risk factors for mental health status among COVID-19 patients. The final multivariable model indicated that female patients (OR = 1.89), medical history (OR = 1.92), and monthly income lower than 5,000 Thai baht (OR = 2.09) were associated with developing long COVID symptoms

**Funding:** This work was funded by the ECforDACH, Walailak University (Ref. No number WU-COE-66-15) awarded to Charuai Suwanbamrung. The funders had no role in study design, data collection and analysis, decision to publish, or preparation of the manuscript.

**Competing interests:** The authors have declared that no competing interests exist.

and mental health status (all p<0.01). This study provides valuable insights into the potential long-term effects of COVID-19 on mental health and enhances understanding of the mechanisms underlying the condition for predicting the occurrence of mental health issues in Thai COVID-19 patients.

## Introduction

The coronavirus disease 2019 (COVID-19) caused by the severe acute respiratory syndrome coronavirus 2 (SARS-CoV-2) has devastated the world's population. While most COVID-19 patients reported experiencing mild respiratory symptoms [1, 2], severe illness or even death has been recorded a significant number of healthy individuals, particularly the elderly or those with specific underlying medical conditions [3]. Recent research has investigated a correlation between mental health disorders and COVID-19 outcomes, indicating that individuals with pre-existing mood disorders had an increased risk of COVID-19 hospitalization and mortality [4]. Throughout the COVID-19 pandemic, there have been conspicuous occurrences of heightened prevalence rates of psychiatric symptoms among the general population worldwide [5]. Anxiety, depression, and post-traumatic stress disorder (PTSD) symptoms have emerged as the most prominent and impactful mental health outcomes [6]. Furthermore, while some COVID-19 cases result in complete symptom resolution, others develop post-COVID-19 conditions (also known as long COVID), characterized by new or persistent symptoms such as fatigue, shortness of breath, and cognitive dysfunction emerging around three months after the acute phase of SARS-CoV-2 infection. Such symptoms typically persist for at least two months with no other explanation by any alternative diagnoses [7]. According to a recent report by WHO, approximately 10% to 20% of SRS-CoV-2 patients may exhibit long COVID symptoms [8].

Governments have been suggested to implement policy measures involving the mental health community and representatives of vulnerable communities during the COVID-19 pandemic [9]. Identifying the risk factors associated with mental health status and post-COVID-19 conditions is crucial in formulating appropriate strategies to minimize the likelihood of COVID-19 patients developing long COVID symptoms and mental health status. However, despite many previous studies on the risk factors of mental health status and long COVID among COVID-19 patients, the findings still remain a vague conclusion, warranting further investigation. A recent study found that several certain demographic groups, including women, the elderly, individuals with chronic illnesses, migrant workers, and students, are more vulnerable to developing psychiatric symptoms than the general population [6]. Acquiring accurate health information and perceiving the pandemic's impact can contribute to adverse mental health outcomes among COVID-19 patients [10]. Conversely, a study conducted in seven Asian middle-income countries revealed that age under 30, a high educational background, being single or separated, and contact with COVID-19 patients are risk factors for mental health issues during the pandemic [11]. Regarding long COVID symptoms, several studies have indicated that advanced age and obesity elevate the risk of developing long COVID [11–14]. In contrast, a cross-sectional study found a higher of long COVID among underweight individuals and younger COVID-19 patients [15]. Moreover, a recent study highlighted that psychological distress before a COVID-19 infection was more strongly linked to the development of long COVID than physical health risk factors such as older age, obesity, asthma, and hypertension [16].

In Thailand, the cumulative number of COVID-19 cases reported until April was 4,728,967. From January to April 2023, the total number of cases was recorded at around 5,000 [17]. A recent report conducted in Thailand indicated that the majority of respondents experienced depression, anxiety, and stress (>87.7%) among recovered COVID-19 patients during the pandemic [18]. Despite several recent studies on long COVID and mental health disorders, these were primarily review papers that relied on highly heterogeneous studies encompassing different questionnaires, time points, countries, and age groups [19], or were based on hospitalized individuals [20]. The results may not accurately represent the experiences of most individuals affected by COVID-19. Hence, our study aims to (1) examine the prevalence of long COVID and mental health status among Thai adults who have recovered from COVID-19, (2) identify the association between mental health issues such as depression, anxiety and stress and long COVID symptoms among COVID-19 participants, and (3) investigate the risk factors associated with the correlation between mental health outcomes and the onset of long COVID in adult patients who have previously contracted COVID-19. By shedding light on the impact of mental health on the development of long COVID among Thai individuals, our study offers valuable insights into the contextual information and associated factors concerning mental health status and post-COVID-19 conditions. This contribution is instrumental in developing effective intervention strategies to reduce the risk of long COVID symptoms and address mental health concerns within the Thai population affected by COVID-19.

## Materials and methods

### Ethics statement

The study protocol adhered to the principles outlined in the Declaration of Helsinki and received approval from the Human Research Ethics Committee of Walailak University (WUEC-22-315-01). Data collection was authorized by the directors of the secondary care hospital and three field hospitals. All participants were provided with a clear explanation of the study's objectives and were assured that their data would be kept anonymous, confidential, and solely used for scientific purposes. Before the interview, oral informed consent was obtained from each participant.

### Study design and participants

The study was a cross-sectional investigation conducted in November 2022 within the community setting of nine subdistricts in Sichon district of Nakhon Si Thammarat province, southern Thailand. Out of the 10,336 individuals diagnosed with COVID-19 between January 2021 and May 2022 in the databases of a secondary care hospital and three field hospitals, eligible participants included those over 18 years old and had no prior mental health disorder diagnosis by a psychiatrist before contracting COVID-19. Ultimately, a list of 9,396 individuals was considered suitable for participation in our research. The target sample size was determined using a web-based sample size calculation tool (http://www.winepi.net). The calculation was based on a reported prevalence of 57% of COVID-19 survivors experiencing long COVID [21], a population size (N) of 9,396 from the hospital databases, a margin of error (d) of 3%, and a confidence interval of 95%. According to the calculation, 942 participants were needed for the study. Proportional allocation using stratified sampling was used to randomly select participants in each subdistrict.

## Study instruments

The study involved administering a structured questionnaire to the participants, which consisted of three sections. The first section aimed to collect socio-demographic information, such as age, gender, education level, marital status, occupation, and monthly income. Additional data were also collected on height, weight, underlying diseases of all participants. Body mass index (BMI) was calculated using the weight to height squared ($kg/m^2$) ratio, and individuals were categorized as underweight, normal weight, overweight at risk, or obese, based on their BMI values ($<18.5$, $18.5$–$22.9$, $23.0$–$24.9$, and $\geq 25.0$, respectively), according to WHO guidelines [22].

The second section of the questionnaire consisted of 13 commonly reported symptoms associated with long COVID, identified through a literature review of a population-based survey on long COVID [14], the WHO case definition of long COVID established through the Delphi consensus [7], and a systemic review of long COVID symptoms [23]. These symptoms included fatigue, shortness of breath, chest tightness, palpitations, cough, amnesia, insomnia, joint pain, muscle pain, asthenia, significant hair loss, headache, and dizziness. The participants were asked "Yes/No" questions to indicate whether they had experienced these symptoms for a period of two months after three months infected with SARS-CoV-2. Those who reported experiencing at least one symptom were classified as having long COVID.

The third section involved the use of the 21-item Depression Anxiety and Stress Scale (DASS-21) developed by Lovibond and Lovibond (1995) [24], which was translated into Thai [18]. The DASS-21 has been widely validated and applied in numerous studies worldwide [25–29], to assess the emotional states of the participants in relation to the three mental health status: depression (7 items), anxiety (7 items), and stress (7 items). Each term was rated on a 4-point scale, ranging from "did not apply to me at all" (0 points), "applied to me some degree or some of the time" (1 point), "applied to me to a considerable degree or a good part of the time" (2 points), and "applied to me very much or most of the time" (3 points). Final scores for each mental health symptom were calculated by adding relevant items and multiplying them by two. The severity levels were categorized as follows: depression (normal: 0–9; mild: 10–13; moderate: 14–20; severe: 21–27; extremely severe: $\geq 28$); anxiety (normal: 0–9; mild: 8–9; moderate: 10–14; severe: 15–19; extremely severe: $\geq 20$), and stress (normal: 0–14; mild: 15–18; moderate: 19–25; severe: 26–33; extremely severe: $\geq 34$). Participants were then categorized based on their scores as "normal", "mild", "moderate", "severe", and "extreme severe" for each symptom. Participants who scored in the "mild" to "extremely severe" range were considered to have mental health symptoms.

To ensure the content validity of the questionnaire, three public health experts evaluated it, and an average index of item-objective congruence score of 0.90 was obtained. None of the items scored lower than the minimum acceptable value of 0.75 was obtained [30]. Cronbach's α was used to evaluate the internal consistency of the questionnaire [31], and the values obtained were 0.83 for the second section on long COVID symptoms and 0.95 for the third section on DASS-21 in the Thai version. Values greater than 0.70 indicate acceptable reliability [32].

## Data collection

Data was collected by a team of 14 village health volunteers (VHVs), who underwent extensive training in conducting health surveys and campaigns to prevent and control infectious diseases. The VHVs received training on how to protect themselves from exposure and infection with SARS-CoV-2, as well as administering the questionnaire to participants correctly. The VHVs visited participants' homes and requested permission to collect data. Each participant was provided a smartphone equipped with an online survey platform to complete the

questionnaire. The questionnaire took roughly 20 minutes to complete, and participants could consult with the VHVs if they had any questions during the survey. BMI measurements of participants were obtained using a digital weight scale carried by VHVs during visits to participants' houses to determine their weight. At the same time, the height values mentioned in Thai identity cards were utilized for height measurements.

## Data analyses

We calculated a standardized score for each COVID symptom reported by study participants based on their frequency and percentage of occurrence. This score was calculated by multiplying the rank of each symptom by its share in each participant's observations, which totalled 100%. For example, if a participant mentioned 3 out of 13 symptoms, such as fatigue, shortness of breath, and chest tightness, each symptom would receive a standardized score of 33.3%.

The association between socio-demographic factors and long COVID among participants was examined using the Chi-squared test and Fisher's Exact test. In addition, we hypothesized that the mental health conditions of COVID-19 patients were likely to be associated with the symptoms of long COVID. To test this hypothesis, we used the odds ratio (OR) to examine the association between depression, anxiety and stress and each symptom of long COVID.

Logistic regression was utilized to identify risk factors associated with mental health status and long COVID among COVID-19 patients. In order to perform logistic regression analysis, the two binary variables of mental health status and long COVID were merged into a new variable with four levels. A binary outcome variable was created for logistic regression modelling, where observations with experience of both mental health status and long COVID were categorized as "1", while the three other levels of the new variable were considered "0". The explanatory variables investigated included gender, age, marital status, education, occupation, monthly income, BMI, and medical history. Univariable models were initially screened for all explanatory variables, and those with $P < 0.20$ were selected as candidates for the final model [33, 34]. The multivariable model consisted of variables with $P < 0.05$ and was used to identify significant risk factors associated with mental health status and long COVID in COVID-19 patients. Interactions between all pairs of explanatory variables were examined to determine any potential confounding effects between the explanatory variables. All statistical analyses were conducted using the R statistical software, with the "stats" package used for building the logistic regression models and the "ggplot2" package used for data visualizations.

## Results

### Characteristics of study participants

Out of the initial 942 participants, data from 939 individuals were collected and included for further analysis. Missing data was found in three respondents. However, after evaluating the study power, excluding their data maintained a statistical power (99%). As a result, data from the remaining 939 participants were used for subsequent analyses. Most participants were female (77.4%) and younger than 60 (84.7%). Approximately 69.1% of participants were married, and around 80% reported having a high school education or lower. The most common occupation among participants was self-employment (60.7%), and the majority (90%) had a monthly income of less than 15,000 Thai baht ($\sim$450 USD). Regarding BMI, 62.5% of participants were overweight, and 41.9% were classified as obese. Among the recorded historical diseases, hypertension was the most prevalent (18.2%), followed by diabetes (11.7%), dyslipidemia (3.6%), cardiovascular disease (2.8%), allergies (1.5%), and other diseases such as thyroid, psoriasis, asthma, and cancer accounted for less than 1% (Table 1).

**Table 1. Descriptive characteristics of study participants with and without long COVID.**

| Characteristics | Total (%) (n = 939) | Participants with long COVID (%) (n = 745) | Participants without long COVID (%) (n = 194) | p-value ($\chi^2$) |
|---|---|---|---|---|
| Gender | | | | < 0.001 |
| Male | 212 (22.6) | 147 (19.7) | 65 (33.5) | |
| Female | 727 (77.4) | 598 (80.3) | 129 (66.5) | |
| Age (year) | | | | 0.867 |
| 19–59 | 795 (84.7) | 632 (84.8) | 163 (84.0) | |
| ≥ 60 | 144 (15.3) | 113 (15.2) | 31 (16.0) | |
| Marital status | | | | 0.021 |
| Single | 210 (22.4) | 153 (20.5) | 57 (29.4) | |
| Currently married | 649 (69.1) | 524 (70.3) | 125 (64.4) | |
| Widowed/separated | 80 (8.5) | 68 (9.1) | 12 (6.2) | |
| Education | | | | 0.221[†] |
| Lower primary | 33 (3.5) | 28 (3.8) | 5 (2.6) | |
| Primary school | 326 (34.7) | 268 (36.0) | 58 (29.9) | |
| High school | 396 (42.2) | 311 (41.7) | 85 (43.8) | |
| Tertiary education | 184 (19.6) | 138 (18.5) | 46 (23.7) | |
| Occupation | | | | 0.168 |
| Agricultural worker | 180 (19.2) | 153 (20.5) | 27 (13.9) | |
| Self-employed worker | 570 (60.7) | 446 (59.9) | 124 (63.9) | |
| Government employee | 41 (4.4) | 33 (4.4) | 8 (4.1) | |
| Private employee | 28 (3.0) | 19 (2.5) | 9 (4.6) | |
| Unemployed worker | 81 (8.6) | 66 (8.9) | 15 (7.7) | |
| Student | 39 (4.2) | 28 (3.8) | 11 (5.7) | |
| Monthly income (Thai baht) | | | | 0.009 |
| ≤ 5,000 | 211 (22.5) | 183 (24.6) | 28 (14.4) | |
| 5,001–10,000 | 422 (44.9) | 335 (45.0) | 87 (44.8) | |
| 10,001–15,000 | 200 (21.3) | 145 (19.5) | 55 (28.4) | |
| 15,001–20,000 | 65 (6.9) | 49 (6.6) | 16 (8.2) | |
| > 20,000 | 41 (4.4) | 33 (4.4) | 8 (4.1) | |
| BMI | | | | 0.014 |
| Underweight | 60 (6.4) | 44 (5.9) | 16 (8.2) | |
| Normal weight | 293 (31.2) | 217 (29.1) | 76 (39.2) | |
| Overweight (at risk) | 193 (20.6) | 156 (20.9) | 37 (19.1) | |
| Overweight (obese) | 393 (41.9) | 328 (44.0) | 65 (33.5) | |
| Medical history | | | | |
| Hypertension | 171 (18.2) | 143 (19.2) | 28 (14.4) | 0.154[†] |
| Diabetes | 110 (11.7) | 93 (12.5) | 17 (8.8) | 0.190 |
| Dyslipidemia | 34 (3.6) | 34 (4.6) | 0 (0.0) | < 0.001 |
| Cardiovascular disease | 26 (2.8) | 18 (2.4) | 8 (4.1) | 0.296 |
| Allergy | 14 (1.5) | 14 (1.9) | 0 (0.0) | 0.087[†] |
| Any diseases | 304 (32.4) | 264 (35.4) | 40 (20.6) | < 0.001 |

[†] Fisher's Exact test

Out of the total participants, 745 (79.3%) reported experiencing prolonged COVID-19 symptoms lasting more than two months, while 194 participants (20.7%) did not report any history of long COVID symptoms. Descriptive analyses revealed significant associations between long COVID conditions and gender ($\chi$2 test, p < 0.001), marital status ($\chi$2 test,

p = 0.021), monthly income (χ2 test, p = 0.009), BMI (χ2 test, p = 0.014), and any historical diseases (χ2 test, p < 0.001). These findings indicate that these factors significantly contribute to developing long COVID (all p < 0.05) (Table 1).

## The characteristics of long COVID symptoms among participants

All thirteen symptoms were reported by 745 participants who experienced long COVID symptoms. The median number of symptoms reported was 4, with an interquartile range (IQR) of one to seven. Among COVID-19 patients with long COVID, more than half of the participants experienced fatigue (72.9%, SE ± 0.02), cough (66.0%, SE ± 0.02), and muscle pain (54.1%, SE ± 0.02), with median standardized scores of 10.0 [IQR 0–16.7], 8.3 [IQR—16.7], and 0.0 [IQR 0–12.5], respectively (Fig 1). The remaining symptoms of long COVID were less frequently reported, with median standardized scores of zero. The frequency of these symptoms was as follows: insomnia (49.4%, SE ± 0.02), headache (48.7%, SE ± 0.02), joint pain (45.0%, SE ± 0.02), shortness of breath (43.5%, SE ± 0.02), dizziness (41.7%, SE ± 0.02), amnesia (41.2%, SE ± 0.02), hair loss (29.7%, SE ± 0.02), palpitations (24.8%, SE ± 0.02), chest tightness (15.3%, SE ± 0.01), and asthenia (12.8%, SE ± 0.01).

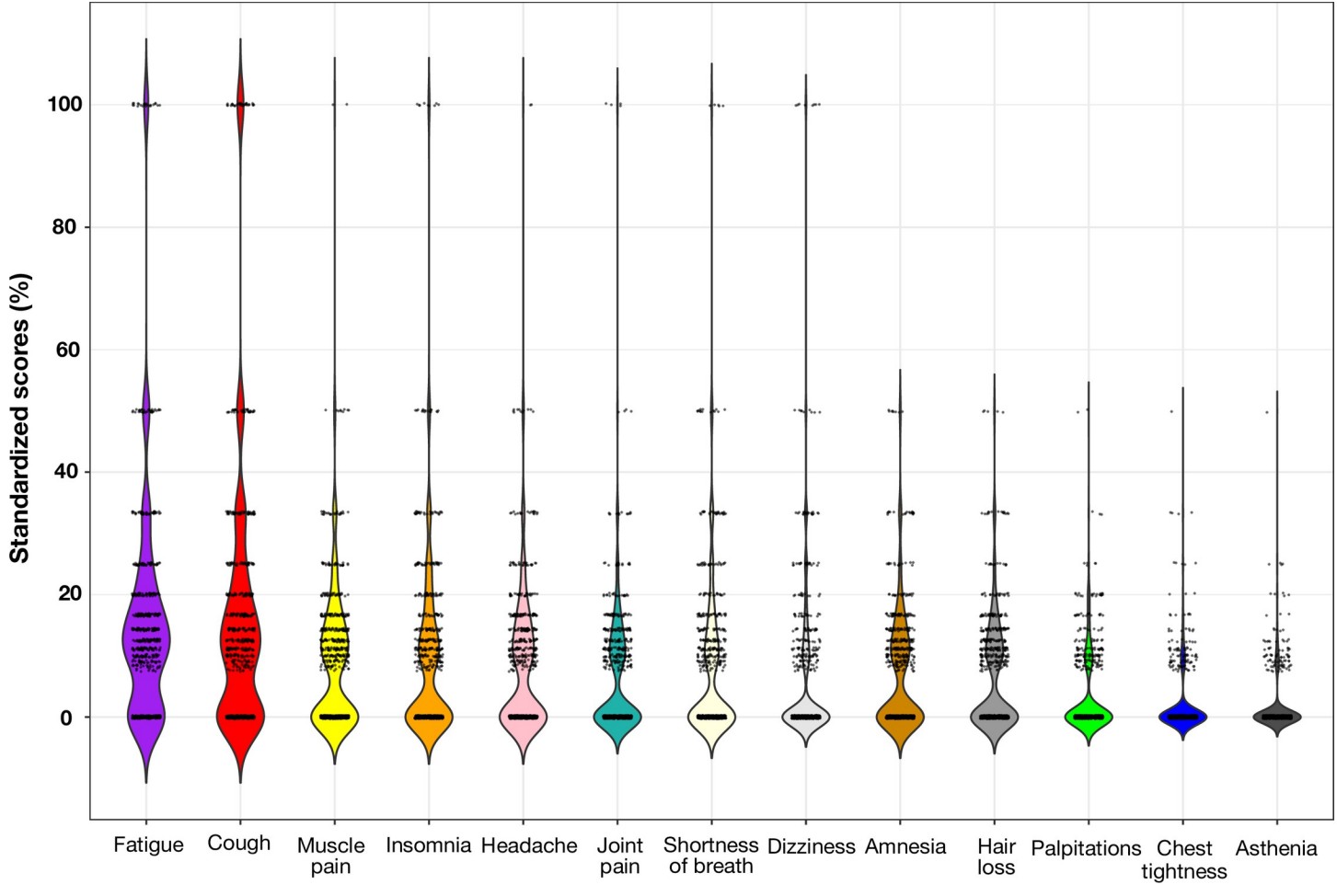

**Fig 1. Standardized scores for each symptom characterized as long COVID listed by participants.**

## Depression, anxiety and stress among recovered COVID-19 participants

Out of 939 participants, 104 (11.1%) were found to have depression, with severity ranging from mild (35 participants) to extremely severe (4 participants). For anxiety, 179 (19.1%) of participants were diagnosed with varying levels of severity, including mild (42 participants), moderate (111 participants), severe (15 participants), and extremely severe (11 participants). Stress was identified in only 42 participants (4.5%), mostly at a mild level (25 participants), followed by moderate (11 participants), severe (5 participants), and extremely severe (1 participant). Among those who had recovered from COVID-19, 33 (3.5%) were diagnosed with all three mental health symptoms of depression, anxiety, and stress. There were 70 (7.5%) participants had two out of the three mental health conditions, and 189 (20.1%) had at least one of the three mental health issues. The levels of depression, anxiety, and stress among COVID-19 participants was illustrated in Table 2.

## Association between mental health status with long COVID among recovered COVID-19 participants

In our survey, we examined the association between all 13 of long COVID symptoms and mental health issues, specifically depression, anxiety, and stress in COVID-19 patients. The study findings revealed that all long COVID symptoms were significantly associated with mental health problems in COVID-19 patients. Participants who experienced any of the long COVID symptoms were at a 4.00-fold higher risk of depression, a 6.93-fold higher risk of anxiety, and a 5.45-fold higher risk of stress (all $p < 0.05$). The symptoms with the highest risk of depression were shortness of breath (OR = 5.21) and chest tightness (OR = 4.33) (all $p < 0.01$). The highest risk factors for anxiety were shortness of breath (OR = 6.98), fatigue (OR = 6.64), and asthenia (OR = 6.16) (all $p < 0.001$). Similarly, for stress, the highest risk factors were shortness of breath (OR = 5.81), fatigue (OR = 4.62), and chest tightness (OR = 3.98) (all $p < 0.01$). Table 3 provides further details on the association between depression, anxiety, and stress status with all long COVID symptoms.

## Risk factor analyses

In the univariable models, seven variables, including gender, marital status, education, occupation, monthly income, BMI and medical history, were identified as potential factors for the multivariable models. However, only three variables were significant in the multivariable model, as marital status, education, occupation, and BMI became non-significant. The results showed that female patients with a medical history had 1.89 times and 1.92 times higher risk of developing long COVID symptoms and experiencing depression, anxiety, and stress, respectively, among COVID-19 patients (all $p < 0.001$). Additionally, patients earning less than 5,000 Thai baht per month had a 2.09 times higher risk of developing long COVID symptoms

**Table 2. The levels of depression, anxiety and stress among recovered COVID-19 patients.**

| Levels of Depression, Anxiety, and Stress | Number of participant (n = 939) (%) | | |
|---|---|---|---|
| | **Depression** | **Anxiety** | **Stress** |
| Normal | 835 (88.9) | 760 (84.1) | 897 (95.5) |
| Mild | 35 (3.7) | 42 (4.4) | 25 (2.7) |
| Moderate | 62 (6.6) | 111 (11.8) | 11 (1.2) |
| Severe | 3 (0.3) | 15 (1.6) | 5 (0.5) |
| Extremely severe | 4 (0.4) | 11 (1.2) | 1 (0.1) |
| **Rate from mild to extremely severe** | 104 (11.1) | 179 (19.1) | 42 (4.5) |

**Table 3. The association between depression, anxiety and stress with and without long COVID among recovered COVID-19 patients (n = 939).**

| Long COVID symptoms | | Depression (n=104) | | | Anxiety (n = 179) | | | Stress (n = 42) | | | Any mental health disorder (n= 189) | | |
|---|---|---|---|---|---|---|---|---|---|---|---|---|---|
| | | Yes | No | ORs | Yes | No | ORs | Yes | No | ORs | Yes | No | ORs |
| Fatigue (n= 543) | Yes | 86 | 457 | 3.95[***] | 156 | 387 | 6.54[***] | 36 | 507 | 4.62[***] | 163 | 380 | 6.10[***] |
| | No | 18 | 378 | | 23 | 373 | | 6 | 390 | | 26 | 370 | |
| Cough (n = 492) | Yes | 69 | 423 | 1.92[**] | 119 | 373 | 2.06[***] | 29 | 463 | 2.09[*] | 127 | 365 | 2.16[***] |
| | No | 35 | 412 | | 60 | 387 | | 13 | 434 | | 62 | 385 | |
| Muscle pain (n = 403) | Yes | 67 | 336 | 2.69[***] | 131 | 272 | 4.90[***] | 28 | 375 | 2.78[**] | 135 | 268 | 4.50[***] |
| | No | 37 | 499 | | 48 | 488 | | 14 | 522 | | 54 | 482 | |
| Insomnia (n = 368) | Yes | 54 | 314 | 1.79[**] | 104 | 264 | 2.61[***] | 17 | 351 | 1.07 | 107 | 261 | 2.44[***] |
| | No | 50 | 521 | | 75 | 496 | | 25 | 546 | | 82 | 489 | |
| Headache (n = 363) | Yes | 70 | 293 | 3.81[***] | 116 | 247 | 3.82[***] | 30 | 333 | 4.23[***] | 118 | 245 | 3.42[***] |
| | No | 34 | 542 | | 63 | 513 | | 12 | 564 | | 71 | 505 | |
| Joint pain (n = 335) | Yes | 64 | 271 | 3.33[***] | 114 | 221 | 4.28[***] | 26 | 309 | 3.09[***] | 118 | 217 | 4.08[***] |
| | No | 40 | 564 | | 65 | 539 | | 16 | 588 | | 71 | 533 | |
| Shortness of breath (n = 324) | Yes | 72 | 252 | 5.21[***] | 127 | 197 | 6.98[***] | 31 | 293 | 5.81[***] | 132 | 192 | 6.73[***] |
| | No | 32 | 538 | | 52 | 563 | | 11 | 604 | | 57 | 558 | |
| Dizziness (n = 311) | Yes | 58 | 253 | 2.90[***] | 97 | 214 | 3.02[***] | 24 | 287 | 2.83[**] | 101 | 210 | 2.95[***] |
| | No | 46 | 582 | | 82 | 546 | | 18 | 610 | | 88 | 540 | |
| Amnesia (n = 307) | Yes | 48 | 259 | 1.91[**] | 103 | 204 | 3.69[***] | 22 | 285 | 2.36[**] | 103 | 204 | 3.21[***] |
| | No | 56 | 576 | | 76 | 556 | | 20 | 612 | | 86 | 546 | |
| Hair loss (n = 221) | Yes | 40 | 181 | 2.26[***] | 72 | 149 | 2.76[***] | 12 | 209 | 1.31 | 75 | 146 | 2.72[***] |
| | No | 64 | 654 | | 107 | 611 | | 30 | 688 | | 114 | 604 | |
| Palpitations (n = 185) | Yes | 42 | 143 | 3.28[***] | 83 | 102 | 5.58[***] | 18 | 167 | 3.28[***] | 85 | 100 | 5.31[***] |
| | No | 62 | 692 | | 96 | 658 | | 24 | 730 | | 104 | 650 | |
| Chest tightness (n = 114) | Yes | 33 | 81 | 4.33[**] | 57 | 57 | 5.76[***] | 14 | 100 | 3.98[***] | 59 | 55 | 5.74[***] |
| | No | 71 | 754 | | 122 | 703 | | 28 | 797 | | 130 | 695 | |
| Asthenia (n = 95) | Yes | 25 | 70 | 3.46[***] | 50 | 45 | 6.16[***] | 10 | 85 | 2.99[**] | 50 | 45 | 5.64[***] |
| | No | 79 | 765 | | 129 | 715 | | 32 | 812 | | 139 | 705 | |
| Any symptom (n = 745) | Yes | 97 | 648 | 4.00[***] | 171 | 574 | 6.93[***] | 40 | 705 | 5.45[*] | 181 | 564 | 7.46[***] |
| | No | 7 | 187 | | 8 | 186 | | 2 | 192 | | 8 | 186 | |

[*] p < 0.05
[**] p < 0.01
[***] p < 0.001

and mental health issues compared to those earning around 10,001–15,000 baht monthly. No significant interactions were found between potential explanatory variables. The statistical models investigating the factors associated with mental health status and long COVID among COVID-19 patients are presented in Table 4.

## Discussion

This study presents novel findings on the prevalence and risk factors associated with mental health issues in COVID-19 patients experiencing long COVID symptoms in southern Thailand. Findings from our study revealed a low prevalence of depression, anxiety, and stress among COVID-19 participants. However, a high prevalence of long COVID symptoms was observed, with fatigue, cough and muscle pain being the most common. The study also identifies shortness of breath, fatigue, and chest tightness as the highest risk factors for mental health

**Table 4. Risk factors associated with long COVID and mental health status among recovered COVID-19 patients.**

| Factors | Univariable models | | | Multivariable model[*] | | |
|---|---|---|---|---|---|---|
| | OR | 95% CI | p-value | OR | 95% CI | p-value |
| **Gender (baseline = male)** | | | | | | |
| Female | 2.05 | 1.44–2.90 | < 0.001 | 1.89 | 1.32–2.69 | < 0.001 |
| Marital status (baseline = single) | | | | | | |
| Currently married | 1.56 | 1.08–2.23 | 0.016 | | | |
| Widowed/separated | 2.11 | 1.10–4.36 | 0.033 | | | |
| Education (baseline = Upper high school) | | | | | | |
| High school | 1.87 | 0.08–1.83 | 0.344 | | | |
| Lower primary | 1.54 | 0.74–5.74 | 0.225 | | | |
| Primary school | 1.22 | 0.99–2.38 | 0.053 | | | |
| Occupation (baseline = Private employee) | | | | | | |
| Student | 1.21 | 0.41–3.48 | 0.728 | | | |
| Agricultural worker | 2.68 | 1.06–6.44 | 0.030 | | | |
| Government employee | 1.95 | 0.64–6.04 | 0.235 | | | |
| Self-employed worker | 1.70 | 0.72–3.76 | 0.201 | | | |
| Unemployed worker | 2.08 | 0.77–5.47 | 0.138 | | | |
| Monthly income (Thai baht) (baseline = 10,001–15,000) | | | | | | |
| ≤ 5,000 | 2.48 | 151–4.15 | < 0.001 | 2.09 | 1.26–3.53 | 0.005 |
| 5,001–10,000 | 1.46 | 0.99–2.15 | 0.057 | 1.38 | 0.92–2.04 | 0.116 |
| 15,001–20,000 | 1.16 | 0.61–2.26 | 0.648 | 1.05 | 0.55–2.07 | 0.878 |
| > 20,000 | 1.57 | 0.71–3.83 | 0.292 | 1.34 | 0.60–2.85 | 0.493 |
| BMI (baseline = Underweight) | | | | | | |
| Normal weight | 1.04 | 0.54–1.92 | 0.907 | | | |
| Overweight (at risk) | 1.53 | 0.77–2.98 | 0.215 | | | |
| Overweight (obese) | 1.84 | 0.95–3.39 | 0.059 | | | |
| Medical history (baseline = Any diseases–No) | | | | | | |
| Any disease–Yes | 2.11 | 1.46–3.12 | < 0.001 | 1.92 | 1.32–2.85 | < 0.001 |

*Model intercept: 0.3933, SE = 0.2

status among COVID-19 patients who experience such issues. Finally, the study finds that female patients, medical history of COVID-19 patients, and low income are associated with the development of long COVID symptoms and mental health status among the study participants.

The COVID-19 pandemic continues to affect communities worldwide, with post-COVID conditions emerging as a significant concern, particularly mental health issues. Prior research conducted in Thailand has demonstrated high levels of depression, anxiety, and stress among COVID-19 patients [18, 35]. However, our study observed a lower prevalence of significant mental health problems among most COVID-19 patients. This lower prevalence may be due to the selection of participants from field hospitals, whereas previous studies collected COVID-19 patients from hospitals or high-risk districts. Furthermore, the study was conducted in November 2022, after lifting strict COVID-19 restrictions, making life in rural communities more manageable. Another possible explanation for the lower prevalence of mental health status among COVID-19 patients in Thailand could be attributed to the healthcare system's increased attention to mental health issues [36, 37]. Recent study has reported that mental health resources and services (i.e., new counselling service—NCS, Psychological Services International—PSI) has become more available and accessible for social support and resilience

of COVID-19 patients in Thailand [38]. This has effectively contributed to reducing mental health status among COVID-19 patients. However, the long-term effects of mental health issues and the need for further monitoring and research on mental health among COVID-19 patients should not be overlooked.

The combination of symptoms experienced by COVID-19 patients with long COVID can vary. Fatigue, shortness of breath, chest pain, joint/muscle pain, headache, insomnia, and loss of smell/taste are among the most commonly reported symptoms [39, 40]. Other symptoms, such as heart palpitations, dizziness, and gastrointestinal issues such as nausea, diarrhoea, and abdominal pain, have also been reported [41–43]. Our study's findings are consistent with these observations, with a high prevalence of long COVID symptoms reported among COVID-19 patients, particularly fatigue, cough, and muscle pain. Long COVID symptoms can persist for several weeks or even months after the initial infection, and their severity can vary considerably between individuals [23]. Long COVID symptoms are thought to be caused by an excessive immune reaction, in which the body's tissues are attacked even after the virus has been eradicated [44]. Additionally, the virus may remain in some individuals, leading to ongoing symptoms [45]. The underlying mechanisms causing the development of long COVID symptoms are not fully understood and require further research.

Our study also found that long COVID symptoms are considered risk factors for developing mental health symptoms among COVID-19 patients. This finding is consistent with recent studies that have linked long COVID symptoms, such as fatigue, shortness of breath, insomnia, and chest tightness, to a higher risk of depression, anxiety, and stress [46, 47]. The distress and interference with daily life caused by long COVID symptoms may contribute to the development of mental health problems [48], along with prolonged illness and uncertainty about recovery leading to fear and frustration [49]. Additionally, the neurological effects of long COVID may also contribute to mental health issues [50]. Recent research has highlighted the efficacy of internet-based cognitive behavioural therapy (CBT) as a non-pharmacological approach for enhancing mental well-being and managing psychiatric patients [51]. Specifically, digital CBT has proven effective in addressing sleep-related issues and improving sleep quality. Insomnia is a prevalent health concern that can contribute to developing psychiatric conditions, including depression and anxiety. Therefore, internet-based CBT represents a promising treatment option for psychiatric symptoms [52]. Internet-based interventions like I-CBT provide convenient therapy access without being limited by geographical distance or scheduling constraints [53]. This is particularly necessary as people have adopted new communication and work patterns during the COVID-19 pandemic.

Our study revealed that female COVID-19 patients face a greater risk of developing long COVID and experiencing mental health issues than male patients. This finding aligns with a recent study, which reported that female COVID-19 patients are 3.3 times more likely to experience long COVID than their counterparts [54]. The investigation also found that females tend to have a more robust immune response to viral infections, generating higher levels of IgG antibodies than males, which may contribute to a more significant response and increase the risk of long COVID symptoms [55]. Furthermore, in Asian societies, women often shoulder a greater burden of domestic responsibilities and caregiving for family members, which can lead to higher levels of stress and psychological distress [56]. The combination of biological, social, and cultural factors may contribute to the observed association between the female gender and an increased risk of long COVID and mental health problems among COVID-19 patients.

In our study, we found that individuals with a history of COVID-19 infection are more likely to develop long COVID and mental health symptoms. Such individuals may experience anxiety or depression due to fear of reinfection or the persistence of symptoms [57]. Additionally, their weakened immune systems make them more susceptible to long-term health

complications, including mental health conditions [49]. Patients with a prior history of mental illnesses are also at an increased risk of developing mental health issues, with more than three times the risk compared to those without such a history [58]. These findings highlight the importance of considering COVID-19 patients' medical history when evaluating the potential for long COVID and mental health disorders.

Moreover, our research indicates that individuals with lower incomes face a higher risk of developing mental health issues and long COVID. Financial stress related to job loss or low income may contribute to the development of mental health issues [59], and limited access to healthcare services could lead to more severe COVID-19 illness and a higher likelihood of experiencing long COVID symptoms [60]. Social and economic factors such as housing conditions and accessibility to healthy food and exercise opportunities may also contribute to mental health problems among COVID-19 patients with lower incomes [61].

Our study has some limitations. First, as mental health problems can have long-lasting effects, cross-sectional studies may not identify the long-term impacts of mental health issues and long COVID symptoms among COVID-19 patients. Thus, follow-up studies using cohort investigations are recommended. Previous research has indicated that the COVID-19 pandemic is associated with hemodynamic alterations in the brain and a decline in olfactory function [62, 63]. Our study used structured questionnaires as the primary approach to evaluate psychiatric symptoms without conducting clinical diagnoses. Thus, the gold standard method for psychiatric diagnoses involving structured clinical interviews and functional neuroimaging is suggested for further investigations [64–66]. Furthermore, our research was conducted within rural communities and relied on hospital databases for data collection. This approach was limited because our survey could not encompass all individuals who recovered from COVID-19 in community settings due to the possibility of numerous cases involving patients with mild symptoms who pursued self-treatment. Additionally, the prevalence and factors associated with long COVID symptoms and mental health status in other areas may differ (i.e., urban and suburban areas) beyond southern Thailand. Although our findings may be generalizable to rural areas, further research is needed to comprehensively understand the prevalence of factors associated with long COVID and mental health issues among COVID-19 patients in various regions of Thailand.

## Conclusion

The COVID-19 pandemic continues to have a significant impact on communities worldwide, with mental health concerns becoming increasingly urgent. Our study has revealed that the prevalence of long COVID, which is associated with mental health conditions, remains alarmingly high and significant. Therefore, it is crucial to conduct further research and closely monitor mental health trends among individuals recovering from COVID-19 to effectively address this pressing issue. Our study provides valuable insights into the potential long-term effects of COVID-19 on mental health, highlights the impact of long COVID on Thai individuals, and improves our understanding of the underlying mechanisms of mental health conditions. These findings can help predict the occurrence of mental health problems in COVID-19 patients in Thailand.

## Supporting information

**S1 Data. Raw data on depression, anxiety, stress and long COVID symptoms were collected from study participants.**
(CSV)

**S1 Checklist. STROBE statement—checklist of items that should be included in reports of observational studies.**
(PDF)

## Acknowledgments

The authors would like to thank the secondary care hospital and three field hospitals in Sichon District, Nakhon Si Thammarat Province. Many thanks to all village health volunteers who provided valuable assistance with data collection. And, the authors would also like to thank all study participants for their participation in the study. Moreover, many thanks to the support from School of Public Health, and ECforDACH, Walailak University.

## Author Contributions

**Conceptualization:** Sarunya Maneerattanasak, Charuai Suwanbamrung.

**Data curation:** Doan Hoang Phu, Sarunya Maneerattanasak, Le Thanh Thao Trang, Muminah Kuning, Aunchalee Like, Hameedah Torpor.

**Formal analysis:** Doan Hoang Phu, Truong Thanh Nam.

**Funding acquisition:** Charuai Suwanbamrung.

**Investigation:** Doan Hoang Phu, Muminah Kuning, Aunchalee Like, Hameedah Torpor, Charuai Suwanbamrung.

**Methodology:** Doan Hoang Phu, Truong Thanh Nam, Charuai Suwanbamrung.

**Project administration:** Charuai Suwanbamrung.

**Resources:** Charuai Suwanbamrung.

**Supervision:** Charuai Suwanbamrung.

**Validation:** Shamarina Shohaimi, Charuai Suwanbamrung.

**Visualization:** Doan Hoang Phu, Le Thanh Thao Trang.

**Writing – original draft:** Doan Hoang Phu, Sarunya Maneerattanasak.

**Writing – review & editing:** Doan Hoang Phu, Sarunya Maneerattanasak, Shamarina Shohaimi, Le Thanh Thao Trang, Truong Thanh Nam, Charuai Suwanbamrung.

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
