## [Decision Letter · Decision Letter 0]

29 May 2023

PONE-D-23-13130Prevalence and Factors Associated with Long COVID and Mental Health Disorders among Recovered COVID-19 Patients in Southern ThailandPLOS ONE

Dear Dr. Suwanbamrung,

Thank you for submitting your manuscript to PLOS ONE. After careful consideration, we feel that it has merit but does not fully meet PLOS ONE’s publication criteria as it currently stands. Therefore, we invite you to submit a revised version of the manuscript that addresses the points raised during the review process.

We look forward to receiving your revised manuscript.

Kind regards,

Md. Saiful Islam, BPH, MPH

Academic Editor

PLOS ONE

Journal Requirements:

"The authors would like to thank the secondary care hospital and three field hospitals in Sichon District, Nakhon Si Thammarat Province. Many thanks to all village health volunteers who provided valuable assistance with data collection. And, the authors would also like to thank all study participants for their participation in the study . Moreover, many thanks to the support from School of Public Health, and ECforDACH, Walailak University."

"This work was funded by the ECforDACH, Walailak University (Ref. No number WU-COE-66-15) awarded to Charuai Suwanbamrung. 

Reviewers' comments:

Reviewer's Responses to Questions

**Comments to the Author**

1. Is the manuscript technically sound, and do the data support the conclusions?

Reviewer #1: Yes

Reviewer #2: Yes

Reviewer #3: Yes

2. Has the statistical analysis been performed appropriately and rigorously? 

Reviewer #1: Yes

Reviewer #2: Yes

Reviewer #3: I Don't Know

3. Have the authors made all data underlying the findings in their manuscript fully available?

Reviewer #1: Yes

Reviewer #2: Yes

Reviewer #3: Yes

4. Is the manuscript presented in an intelligible fashion and written in standard English?

Reviewer #1: Yes

Reviewer #2: Yes

Reviewer #3: Yes

5. Review Comments to the Author

Reviewer #1: I have the followng comments for the authors to address and happy to review this paper again.

1) Under the Introduction, please disucss the followng Key research findings:

Impact of viral respiratory epidemics on mental health

Search PubMed for: The most significant mental health outcomes reported include anxiety, depression, and post-traumatic stress disorder symptoms. The subgroups identified to have a higher risk of psychiatric symptoms among the general public include females, the elderly, individuals with chronic illness, migrant workers, and students

A systematic review of COVID-19 on mental health

Search for PubMed: Relatively high rates of symptoms of anxiety (6.33% to 50.9%), depression (14.6% to 48.3%), post-traumatic stress disorder (7% to 53.8%), psychological distress (34.43% to 38%), and stress (8.1% to 81.9%) are reported in the general population during the COVID-19 pandemic in China, Spain, Italy, Iran, the US, Turkey, Nepal, and Denmark.

The impact of COVID-19 on three continents and its relationship with physical health:

Search for PubMed: The results showed that Poland and the Philippines were the two countries with the highest levels of anxiety, depression and stress; conversely, Vietnam had the lowest mean scores in these areas. Chain mediation model showed the need for health information, and the perceived impact of the pandemic were sequential mediators between physical symptoms resembling COVID-19 infection (predictor) and consequent mental health status (outcome).

The impact of COVID-19 on developing countries:

Search PubMed for: The risk factors for adverse mental health during the COVID-19 pandemic include age <30 years, high education background, single and separated status, discrimination by other countries and contact with people with COVID-19 (p<0.05).

Government response during the pandemic:

Search PubMed for: The overall proportion of study participants with clinically significant depressive symptoms was 21.39% (95% CI 19.37-23.47). Governments that enacted stringent measures to contain the spread of COVID-19 benefited the mental health of their population.

Worst outcome of COVID infection due to depression

Search PubMed for: The results of this systematic review and meta-analysis examining the association between preexisting mood disorders and COVID-19 outcomes suggest that individuals with preexisting mood disorders are at higher risk of COVID-19 hospitalization and death and should be categorized as an at-risk group on the basis of a preexisting condition.

Post COVID and depression

Search PubMed for: Onset and frequency of depression in post-COVID-19 syndrome: Following recovery from COVID-19, an increasing proportion of individuals have reported the persistence and/or new onset of symptoms which collectively have been identified as post-COVID-19 syndrome by the National Institute for Health and Care Excellence. Although depressive symptoms in the acute phase of COVID-19 have been well characterized, the frequency of depression following recovery of the acute phase remains unknown. Herein, we sought to determine the frequency of depressive symptoms and clinically-significant depression more than 12 weeks following SARS-CoV-2 infection.

Search PubMed for “Fatigue and cognitive impairment in Post-COVID-19 Syndrome: Meta-analysis revealed that the proportion of individuals experiencing fatigue 12 or more weeks following COVID-19 diagnosis was 0.32 (95% CI, 0.27, 0.37; p < 0.001; n = 25,268; I2 = 99.1%). A significant proportion of individuals experience persistent fatigue and/or cognitive impairment following resolution of acute COVID-19. The frequency and debilitating nature of the foregoing symptoms provides the impetus to characterize the underlying neurobiological substrates and how to best treat these phenomena.”

2) For the methods, it is important to state DASS-21 is validated in many countries during the COVID-19 pandemic.

China: Immediate Psychological Responses and Associated Factors during the Initial Stage of the 2019 Coronavirus Disease (COVID-19) Epidemic among the General Population in China. Int J Environ Res Public Health. 2020;17(5):1729. Published 2020 Mar 6. doi:10.3390/ijerph17051729

Spain: The Impact of 2019 Coronavirus Disease (COVID-19) Pandemic on Physical and Mental Health: A Comparison between China and Spain. JMIR Form Res. 2021 Apr 22. doi: 10.2196/27818. Epub ahead of print. PMID: 33900933.

The US: The impact of the COVID-19 pandemic on physical and mental health in the two largest economies in the world: a comparison between the United States and China. J Behav Med. 2021 Jun 14:1–19. doi: 10.1007/s10865-021-00237-7. Epub ahead of print. PMID: 34128179; PMCID: PMC8202541.

Poland: The Association Between Physical and Mental Health and Face Mask Use During the COVID-19 Pandemic: A Comparison of Two Countries With Different Views and Practices. Front Psychiatry. 2020;11:569981. Published 2020 Sep 9. doi:10.3389/fpsyt.2020.569981

Iran: https://www.mdpi.com/2673-5318/2/1/6

Philippines: Psychological impact of COVID-19 pandemic in the Philippines. J Affect Disord. 2020 Aug 24;277:379-391. doi: 10.1016/j.jad.2020.08.043. Epub ahead of print. PMID: 32861839.

Vietnan: Evaluating the Psychological Impacts Related to COVID-19 of Vietnamese People Under the First Nationwide Partial lockdown in Vietnam. Front Psychiatry. 2020 Sep 2;11:824. doi: 10.3389/fpsyt.2020.00824. PMID: 32982807; PMCID: PMC7492529.

3) Under discussion, please discuss how internet interventions can improve mental health of general population:

The most evidence-based treatment is cognitive behaviour therapy (CBT), especially Internet CBT that can prevent the spread of infection during the pandemic.

Use of Cognitive Behavior Therapy (CBT) to treat psychiatric symptoms during COVID-19:

Mental Health Strategies to Combat the Psychological Impact of COVID-19 Beyond Paranoia and Panic. Ann Acad Med Singapore. 2020;49(3):155‐160.

Cost-effectiveness of iCBT:

Moodle: The cost effective solution for internet cognitive behavioral therapy (I-CBT) interventions. Technol Health Care. 2017;25(1):163-165. doi: 10.3233/THC-161261. PMID: 27689560.

Internet CBT can treat psychiatric symptoms such as insomnia:

Efficacy of digital cognitive behavioural therapy for insomnia: a meta-analysis of randomised controlled trials. Sleep Med. 2020 Aug 26;75:315-325. doi: 10.1016/j.sleep.2020.08.020. Epub ahead of print. PMID: 32950013.

4) Under discussion, please discuss further research on COVID-19 burnout based on the following finding:

Search PubMed for: Burnout is an important public health issue at times of the COVID-19 pandemic. Current measures which focus on work-based burnout have limitations in length and/or relevance. When stepping into the post-pandemic as a new Norm Era, the burnout scale for the general population is urgently needed to fill the gap. This study aimed to develop a COVID-19 Burnout Views Scale (COVID-19 BVS) to measure burnout views of the general public in a Chinese context and examine its psychometric properties.

5) Please add the following limitation:

The COVID-19 pandemic was found to cause hemodynamic changes in the brain (Olszewska-Guizzo et al 2021) and impairment in olfactory function (Ho et al 2021). This study mainly used self-reported questionnaires to measure psychiatric symptoms and did not make clinical diagnosis. The gold standard for establishing psychiatric diagnosis involved structured clinical interview and functional neuroimaging should be applied in the future face-to-face research after COVID-19 restrictions are removed. (Husain et al 2019, Husain et al 2020, Ho et al 2020).

References:

Olszewska-Guizzo, A.; Mukoyama, A.; Naganawa, S.; Dan, I.; Husain, S.F.; Ho, C.S.; Ho, R. Hemodynamic Response to Three Types of Urban Spaces before and after Lockdown during the COVID-19 Pandemic. Int. J. Environ. Res. Public Health 2021, 18, 6118. https://doi.org/10.3390/ijerph18116118

Ho RC et al Comparison of Brain Activation Patterns during Olfactory Stimuli between Recovered COVID-19 Patients and Healthy Controls: A Functional Near-Infrared Spectroscopy (fNIRS) Study. Brain Sciences. 2021; 11(8):968. https://doi.org/10.3390/brainsci11080968

Husain SF, Yu R, Tang TB, et al. Validating a functional near-infrared spectroscopy diagnostic paradigm for Major Depressive Disorder. Sci Rep. 2020;10(1):9740. Published 2020 Jun 16. doi:10.1038/s41598-020-66784-2

Husain SF, Tang TB, Yu R,et al . Cortical haemodynamic response measured by functional near infrared spectroscopy during a verbal fluency task in patients with major depression and borderline personality disorder. EBioMedicine. 2019 Dec 23;51:102586. doi: 10.1016/j.ebiom.2019.11.047. PMID: 31877417.

Ho CSH, Lim LJH, Lim AQ, et al. Diagnostic and Predictive Applications of Functional Near-Infrared Spectroscopy for Major Depressive Disorder: A Systematic Review. Front Psychiatry. 2020;11:378. Published 2020 May 6. doi:10.3389/fpsyt.2020.00378

Reviewer #2: This study aimed to investigate the relationship and risk factors between long COVID-19 and mental health among adult participants who recovered from COVID-19 in southern Thailand. The study collected basic information such as age, gender, medical history, and current symptoms of COVID-19 and psychological problems such as depression, anxiety, and stress. This study present value findings on the prevalence and risk factors associated with mental health and long COVID-19 in southern Thailand. However, there are some issues that need to be addressed:

The main concern is that combining psychological problems and long-term COVID-19 as outcome indicators may not be reasonable. A separate analysis is suggested. Other suggestions are as below.

1. In instruction, the study did not clearly explain which psychological problems have a significant impact on long COVID-19, nor did it emphasize the importance of studying psychological problems. Also, the study status of this area should be reviewed.

2. It is unclear how participants were "randomly chosen", and the limitations of using hospital databases should be discussed.

3. Specific inclusion and exclusion criteria were not provided, which should be added. Such as whether the participants who had just a single infection were included.

4. The DASS-21, which is a self-reported questionnaire, may just assess the “mental symptoms” and not directly indicate "mental health disorders".

5. Only “any” or “no” for medical history is too sketchy.

6. There is no need to add figure legends in the result section.

7. In discussion, some related studies about COVID-19 and mental health should be compared with the findings in this study.

Reviewer #3: Thank you for providing the opportunity to review this manuscript. The paper presents a potentially exciting study timely on the current problems. This article makes some important contributions in terms of identifying factors associated with long COVID and its association with mental health problems. The results of this study may have important policy implications in its field and may contribute to developing strategies for addressing the issues. However, reading through the paper, I suggest authors improve the manuscript with revisions in some sections to get published.

For the detailed comments, please refer to the document attached.

6. PLOS authors have the option to publish the peer review history of their article (what does this mean?). If published, this will include your full peer review and any attached files.

Reviewer #1: No

Reviewer #2: No

Reviewer #3: **Yes: **Hridaya Raj Devkota

---

## [Author Response · Author response to Decision Letter 0]

26 Jun 2023

Rebuttal Letter

Dear Editor and Reviewers, 

Many thanks to the reviewers for his/her comments and suggestions. Please find below a revised manuscript as well as detailed responses to specific comments:

Reviewer #1: I have the following comments for the authors to address and happy to review this paper again.

1) Under the Introduction, please discuss the following Key research findings:

Impact of viral respiratory epidemics on mental health 

Search PubMed for: The most significant mental health outcomes reported include anxiety, depression, and post-traumatic stress disorder symptoms. The subgroups identified to have a higher risk of psychiatric symptoms among the general public include females, the elderly, individuals with chronic illness, migrant workers, and students. 

Response – We have now added more text and provided citation/s as suggested: 

[Anxiety, depression, and post-traumatic stress disorder (PTSD) symptoms have emerged as the most prominent and impactful mental health outcomes (6)]

[A recent study found that several certain demographic groups, including women, the elderly, individuals with chronic illnesses, migrant workers, and students, are more vulnerable to developing psychiatric symptoms than the general population (6).]

A systematic review of COVID-19 on mental health Search for PubMed: Relatively high rates of symptoms of anxiety (6.33% to 50.9%), depression (14.6% to 48.3%), post-traumatic stress disorder (7% to 53.8%), psychological distress (34.43% to 38%), and stress (8.1% to 81.9%) are reported in the general population during the COVID-19 pandemic in China, Spain, Italy, Iran, the US, Turkey, Nepal, and Denmark.

Response – We have now added more text and provided citation/s as suggested: 

[Throughout the COVID-19 pandemic, there have been conspicuous occurrences of heightened prevalence rates of psychiatric symptoms among the general population worldwide (5).]

The impact of COVID-19 on three continents and its relationship with physical health:

Search for PubMed: The results showed that Poland and the Philippines were the two countries with the highest levels of anxiety, depression and stress; conversely, Vietnam had the lowest mean scores in these areas. Chain mediation model showed the need for health information, and the perceived impact of the pandemic were sequential mediators between physical symptoms resembling COVID-19 infection (predictor) and consequent mental health status (outcome).

Response – We have now added more text and provided citation/s as suggested: 

[Acquiring accurate health information and perceiving the pandemic's impact can contribute to adverse mental health outcomes among COVID-19 patients (10).]

The impact of COVID-19 on developing countries:

Search PubMed for: The risk factors for adverse mental health during the COVID-19 pandemic include age <30 years, high education background, single and separated status, discrimination by other countries and contact with people with COVID-19 (p<0.05).

Response – We have now added more text and provided citation/s as suggested: 

[Conversely, a study conducted in seven Asian middle-income countries revealed that age under 30, a high educational background, being single or separated, and contact with COVID-19 patients are risk factors for mental health issues during the pandemic (11)]

Government response during the pandemic: 

Search PubMed for: The overall proportion of study participants with clinically significant depressive symptoms was 21.39% (95% CI 19.37-23.47). Governments that enacted stringent measures to contain the spread of COVID-19 benefited the mental health of their population.

Response – We have now added more text and provided citation/s as suggested: 

[Governments have been suggested to implement policy measures involving the mental health community and representatives of vulnerable communities during the COVID-19 pandemic (9).]

Worst outcome of COVID infection due to depression

Search PubMed for: The results of this systematic review and meta-analysis examining the association between preexisting mood disorders and COVID-19 outcomes suggest that individuals with preexisting mood disorders are at higher risk of COVID-19 hospitalization and death and should be categorized as an at-risk group on the basis of a preexisting condition.

Response – We have now added more text and provided citation/s as suggested: 

[Recent research has investigated a correlation between mental health disorders and COVID-19 outcomes, indicating that individuals with pre-existing mood disorders had an increased risk of COVID-19 hospitalization and mortality (4).]

Post COVID and depression Search PubMed for: 

Onset and frequency of depression in post-COVID-19 syndrome: Following recovery from COVID-19, an increasing proportion of individuals have reported the persistence and/or new onset of symptoms which collectively have been identified as post-COVID-19 syndrome by the National Institute for Health and Care Excellence. Although depressive symptoms in the acute phase of COVID-19 have been well characterized, the frequency of depression following recovery of the acute phase remains unknown. Herein, we sought to determine the frequency of depressive symptoms and clinically-significant depression more than 12 weeks following SARS-CoV-2 infection.

Response – We have now added more text and provided citation/s as suggested: 

[Despite several recent studies on long COVID and mental health disorders, these were primarily review papers that relied on highly heterogeneous studies encompassing different questionnaires, time points, countries, and age groups (19), or were based on hospitalized individuals (20).]

Search PubMed for “Fatigue and cognitive impairment in Post-COVID-19 Syndrome: Meta-analysis revealed that the proportion of individuals experiencing fatigue 12 or more weeks following COVID-19 diagnosis was 0.32 (95% CI, 0.27, 0.37; p < 0.001; n = 25,268; I2 = 99.1%). A significant proportion of individuals experience persistent fatigue and/or cognitive impairment following resolution of acute COVID-19. The frequency and debilitating nature of the foregoing symptoms provides the impetus to characterize the underlying neurobiological substrates and how to best treat these phenomena.”

Response – We have now added more text and provided citation/s as suggested: 

[Despite several recent studies on long COVID and mental health disorders, these were primarily review papers that relied on highly heterogeneous studies encompassing different questionnaires, time points, countries, and age groups (19), or were based on hospitalized individuals (20).]

2) For the methods, it is important to state DASS-21 is validated in many countries during the COVID-19 pandemic.

China: Immediate Psychological Responses and Associated Factors during the Initial Stage of the 2019 Coronavirus Disease (COVID-19) Epidemic among the General Population in China. Int J Environ Res Public Health. 2020;17(5):1729. Published 2020 Mar 6. doi:10.3390/ijerph17051729

Spain: The Impact of 2019 Coronavirus Disease (COVID-19) Pandemic on Physical and Mental Health: A Comparison between China and Spain. JMIR Form Res. 2021 Apr 22. doi: 10.2196/27818. Epub ahead of print. PMID: 33900933.

The US: The impact of the COVID-19 pandemic on physical and mental health in the two largest economies in the world: a comparison between the United States and China. J Behav Med. 2021 Jun 14:1–19. doi: 10.1007/s10865-021-00237-7. Epub ahead of print. PMID: 34128179; PMCID: PMC8202541.

Poland: The Association Between Physical and Mental Health and Face Mask Use During the COVID-19 Pandemic: A Comparison of Two Countries With Different Views and Practices. Front Psychiatry. 2020;11:569981. Published 2020 Sep 9. doi:10.3389/fpsyt.2020.569981

Iran: https://www.mdpi.com/2673-5318/2/1/6

Philippines: Psychological impact of COVID-19 pandemic in the Philippines. J Affect Disord. 2020 Aug 24;277:379-391. doi: 10.1016/j.jad.2020.08.043. Epub ahead of print. PMID: 32861839.

Vietnam: Evaluating the Psychological Impacts Related to COVID-19 of Vietnamese People Under the First Nationwide Partial lockdown in Vietnam. Front Psychiatry. 2020 Sep 2;11:824. doi: 10.3389/fpsyt.2020.00824. PMID: 32982807; PMCID: PMC7492529.

Response – We have now added more text and provided citation/s as suggested: 

[The DASS-21 has been widely validated and applied in numerous studies worldwide (25–29).]

3) Under discussion, please discuss how internet interventions can improve mental health of general population: 

The most evidence-based treatment is cognitive behaviour therapy (CBT), especially Internet CBT that can prevent the spread of infection during the pandemic.

Use of Cognitive Behavior Therapy (CBT) to treat psychiatric symptoms during COVID-19:

Mental Health Strategies to Combat the Psychological Impact of COVID-19 Beyond Paranoia and Panic. Ann Acad Med Singapore. 2020;49(3):155‐160.

Response – We have now added more text and provided citation/s as suggested: 

[Recent research has highlighted the efficacy of internet-based cognitive behavioural therapy (CBT) as a non-pharmacological approach for enhancing mental well-being and managing psychiatric patients (51).]

Cost-effectiveness of iCBT:

Moodle: The cost-effective solution for internet cognitive behavioral therapy (I-CBT) interventions. Technol Health Care. 2017;25(1):163-165. doi: 10.3233/THC-161261. PMID: 27689560. 

Response – We have now added more text and provided citation/s as suggested: 

[Internet-based interventions like I-CBT provide convenient therapy access without being limited by geographical distance or scheduling constraints (53).]

Internet CBT can treat psychiatric symptoms such as insomnia:

Efficacy of digital cognitive behavioural therapy for insomnia: a meta-analysis of randomised controlled trials. Sleep Med. 2020 Aug 26;75:315-325. doi: 10.1016/j.sleep.2020.08.020. Epub ahead of print. PMID: 32950013.0/0/00 0:00:00 AM0/0/00 0:00:00 AM0/0/00 0:00:00 AM

Response - We have now added more text and provided citation/s as suggested:

[Specifically, digital CBT has proven effective in addressing sleep-related issues and improving sleep quality. Insomnia is a prevalent health concern that can contribute to developing psychiatric conditions, including depression and anxiety. Therefore, internet-based CBT represents a promising treatment option for psychiatric symptoms (52).]

4) Under discussion, please discuss further research on COVID-19 burnout based on the following finding:

Search PubMed for: Burnout is an important public health issue at times of the COVID-19 pandemic. Current measures which focus on work-based burnout have limitations in length and/or relevance. When stepping into the post-pandemic as a new Norm Era, the burnout scale for the general population is urgently needed to fill the gap. This study aimed to develop a COVID-19 Burnout Views Scale (COVID-19 BVS) to measure burnout views of the general public in a Chinese context and examine its psychometric properties.

Response – We have thoroughly examined the suggested article, which presents an interesting study on assessing burnout views among the general population in China during the prolonged pandemic in the Chinese zero-COVID context. Upon careful consideration, we have concluded that delving further into the topic of COVID-19 burnout in our discussion would not directly align with the key findings of our study. Additionally, it is essential to note that our study specifically targeted recovered COVID-19 patients, whereas the suggested research focused on the general public.

5) Please add the following limitation:

The COVID-19 pandemic was found to cause hemodynamic changes in the brain (Olszewska-Guizzo et al 2021) and impairment in olfactory function (Ho et al 2021). This study mainly used self-reported questionnaires to measure psychiatric symptoms and did not make clinical diagnosis. The gold standard for establishing psychiatric diagnosis involved structured clinical interview and functional neuroimaging should be applied in the future face-to-face research after COVID-19 restrictions are removed. (Husain et al 2019, Husain et al 2020, Ho et al 2020).

Response – We have now added more text and provided citation/s as suggested: 

[Previous research has indicated that the COVID-19 pandemic is associated with hemodynamic alterations in the brain and a decline in olfactory function (62,63). Our study used structured questionnaires as the primary approach to evaluate psychiatric symptoms without conducting clinical diagnoses. Thus, the gold standard method for psychiatric diagnoses involving structured clinical interviews and functional neuroimaging is suggested for further investigations (64–66)]

References:

Olszewska-Guizzo, A.; Mukoyama, A.; Naganawa, S.; Dan, I.; Husain, S.F.; Ho, C.S.; Ho, R. Hemodynamic Response to Three Types of Urban Spaces before and after Lockdown during the COVID-19 Pandemic. Int. J. Environ. Res. Public Health 2021, 18, 6118. https://doi.org/10.3390/ijerph18116118

Ho RC et al Comparison of Brain Activation Patterns during Olfactory Stimuli between Recovered COVID-19 Patients and Healthy Controls: A Functional Near-Infrared Spectroscopy (fNIRS) Study. Brain Sciences. 2021; 11(8):968. https://doi.org/10.3390/brainsci11080968

Husain SF, Yu R, Tang TB, et al. Validating a functional near-infrared spectroscopy diagnostic paradigm for Major Depressive Disorder. Sci Rep. 2020;10(1):9740. Published 2020 Jun 16. doi:10.1038/s41598-020-66784-2

Husain SF, Tang TB, Yu R,et al . Cortical haemodynamic response measured by functional near infrared spectroscopy during a verbal fluency task in patients with major depression and borderline personality disorder. EBioMedicine. 2019 Dec 23;51:102586. doi: 10.1016/j.ebiom.2019.11.047. PMID: 31877417.

Ho CSH, Lim LJH, Lim AQ, et al. Diagnostic and Predictive Applications of Functional Near-Infrared Spectroscopy for Major Depressive Disorder: A Systematic Review. Front Psychiatry. 2020;11:378. Published 2020 May 6. doi:10.3389/fpsyt.2020.00378

_ _ _ _ _ o0o _ _ _ _ 

Reviewer #2: This study aimed to investigate the relationship and risk factors between long COVID-19 and mental health among adult participants who recovered from COVID-19 in southern Thailand. The study collected basic information such as age, gender, medical history, and current symptoms of COVID-19 and psychological problems such as depression, anxiety, and stress. This study present value findings on the prevalence and risk factors associated with mental health and long COVID-19 in southern Thailand. However, there are some issues that need to be addressed:

Many thanks to the reviewer for the comments and suggestions. Please find below a revised manuscript as well as detailed responses to specific comments:

The main concern is that combining psychological problems and long-term COVID-19 as outcome indicators may not be reasonable. A separate analysis is suggested. 

Response – We express our gratitude to the reviewer for the comment and suggestions. Our study aimed to explore the prevalence and association between mental health status and long COVID symptoms among individuals affected by COVID-19 and investigate the risk factors associated with mental health status and long COVID conditions among recovered COVID-19 patients. Initially, we had planned to analyze psychological issues and long COVID as separate components using logistic models. However, after examining the association between long COVID and mental health status, we observed that all long COVID symptoms were associated with mental health status. Furthermore, due to the massive number of symptoms involved, with three symptoms for mental health status (depression, anxiety, and stress) and thirteen for long COVID symptoms (fatigue, cough, muscle pain, insomnia, headache, joint pain, shortness of breath, dizziness, amnesia, hair loss, palpitations, chest tightness, asthenia), the results from these separate models were extensive. As a result, we combined the binary variables of mental health status and long COVID into a new variable with four levels and focused our analyses on a subgroup of individuals experiencing both mental health issues and long COVID to identify the risk factors. Findings from our study contribute to advancing our understanding of the interplay between mental health status and long COVID symptoms, shedding light on the risk factors associated with these conditions among individuals who have recovered from COVID-19.

1. In instruction, the study did not clearly explain which psychological problems have a significant impact on long COVID-19, nor did it emphasize the importance of studying psychological problems. Also, the study status of this area should be reviewed.

Response – We have now added more text in the introduction as follows:

[Recent research has investigated a correlation between mental health disorders and COVID-19 outcomes, indicating that individuals with pre-existing mood disorders had an increased risk of COVID-19 hospitalization and mortality (4). Throughout the COVID-19 pandemic, there have been conspicuous occurrences of heightened prevalence rates of psychiatric symptoms among the general population worldwide (5). Anxiety, depression, and post-traumatic stress disorder (PTSD) symptoms have emerged as the most prominent and impactful mental health outcomes (6).]

[However, despite many previous studies on the risk factors of mental health status and long COVID among COVID-19 patients, the findings still remain a vague conclusion, warranting further investigation. A recent study found that several certain demographic groups, including women, the elderly, individuals with chronic illnesses, migrant workers, and students, are more vulnerable to developing psychiatric symptoms than the general population (6). Acquiring accurate health information and perceiving the pandemic's impact can contribute to adverse mental health outcomes among COVID-19 patients (10). Conversely, a study conducted in seven Asian middle-income countries revealed that age under 30, a high educational background, being single or separated, and contact with COVID-19 patients are risk factors for mental health issues during the pandemic (11).]

[A recent report conducted in Thailand indicated that the majority of respondents experienced depression, anxiety, and stress (>87.7%) among recovered COVID-19 patients during the pandemic (18). Despite several recent studies on long COVID and mental health disorders, these were primarily review papers that relied on highly heterogeneous studies encompassing different questionnaires, time points, countries, and age groups (19), or were based on hospitalized individuals (20). The results may not accurately represent the experiences of most individuals affected by COVID-19.]

2. It is unclear how participants were "randomly chosen", and the limitations of using hospital databases should be discussed.

Response – We have now added more informatio on the participant selection in the text. 

[Out of the 10,336 individuals diagnosed with COVID-19 between January 2021 and May 2022 in the databases of a secondary care hospital and three field hospitals, eligible participants included those over 18 years old and had no prior mental health disorder diagnosis by a psychiatrist before contracting COVID-19. Ultimately, a list of 9,396 individuals was considered suitable for participation in our research.]

We have now added more text in the limitation of the disscussion section. 

[Furthermore, our research was conducted within rural communities and relied on hospital databases for data collection. This approach was limited because our survey could not encompass all individuals who recovered from COVID-19 in community settings due to the possibility of numerous cases involving patients with mild symptoms who pursued self-treatment.]

3. Specific inclusion and exclusion criteria were not provided, which should be added. Such as whether the participants who had just a single infection were included.

Response – We have now added more information on the inclusion and exclusion in the text. 

[Eligible participants included those over 18 years old and had no prior mental health disorder diagnosis by a psychiatrist before contracting COVID-19.]

For the information on the number of infections (single or multiple infections of COVID-19), we acknowledge that we inadvertently missed gathering information on the number of COVID-19 infections experienced by individuals during our data collection process. Regrettably, the provided database did not include this specific detail regarding the number of infections of COVID-19 patients. However, this is an intriguing aspect worthy of attention. The number of COVID-19 infections individual experiences could have a notable impact on their mental health and long COVID symptoms. This factor warrants further investigation and should be considered in our future research efforts.

4. The DASS-21, which is a self-reported questionnaire, may just assess the “mental symptoms” and not directly indicate "mental health disorders".

Response – The phrase “mental health disorders” has been now replaced by “mental health symptoms/ mental health status”

5. Only “any” or “no” for medical history is too sketchy.

Response – Thanks for your comment. In our initial approach, we attempted to analyze the medical history of diseases by running separate logistic regression models. The attached table presents the results from the univariable models when the diseases were analyzed individually. However, none of the diseases were identified as risk factors for long-term COVID and mental health status outcome variables. Nevertheless, upon creating a new variable called "any disease," which captured the presence of any reported diseases across all participants, we found that this new variable was a significant risk factor for the outcome variable (Table 3, in the manuscript). 

Factors Univariable models

Medical history (baseline = No) 

Hypertension- Yes 1.41 0.92 - 2.22 0.127

Diabetes- Yes 1.49 0.88 - 2.63 0.153

Cardiovascular disease - Yes 0.58 0.25 - 1.42 0.202

Allergy - Yes nc nc nc

… 

Any disease – Yes 2.11 1.46 – 3.12 < 0.001

nc: data is too small for analysis (only 14/939 participants reported allergy). For other diseases (i.e., Thyroid (n=5), kidney diseases (n=3), Psoriasis (n=4), Asthma (n=6), and cancer (n=6): the number of participants reported these diseases too small for analysis)

6. There is no need to add figure legends in the result section.

Response – We are unsure about this comment since we do not have any figure legends in the result section. In case you mentioned the Figure captions, we followed the guideline for submission of PLoS ONE journal: “Figure captions must be inserted in the text of the manuscript, immediately following the paragraph in which the figure is first cited.”

7. In discussion, some related studies about COVID-19 and mental health should be compared with the findings in this study.

Response – We have had the comparison and interpretation of the findings of our study with previous studies in the discussion as follows:

Prevalence of mental health status: 

[Prior research conducted in Thailand has demonstrated high levels of depression, anxiety, and stress among COVID-19 patients (24,33). However, this study observed a lower prevalence of significant mental health problems among most COVID-19 patients. This lower prevalence may be due to the selection of participants from field hospitals, whereas previous studies collected COVID-19 patients from hospitals or high-risk districts. Furthermore, the study was conducted in November 2022, after lifting strict COVID-19 restrictions, making life in rural communities more manageable. Another possible explanation for the lower prevalence of mental health status among COVID-19 patients in Thailand could be attributed to the healthcare system’s increased attention to mental health issues (34,35). Recent study has reported that mental health resources and services (i.e., new counselling service - NCS, Psychological Services International - PSI) has become more available and accessible for social support and resilience of COVID-19 patients in Thailand (36)]

Prevalence of long COVID symptoms

[The combination of symptoms experienced by COVID-19 patients with long COVID can vary. Fatigue, shortness of breath, chest pain, joint/muscle pain, headache, insomnia, and loss of smell/taste are among the most commonly reported symptoms (37,38). Other symptoms, such as heart palpitations, dizziness, and gastrointestinal issues such as nausea, diarrhoea, and abdominal pain, have also been reported (39–41). Our study's findings are consistent with these observations, with a high prevalence of long COVID symptoms reported among COVID-19 patients, particularly fatigue, cough, and muscle pain]

Long COVID symptoms are considered risk factors for developing mental health symptoms

[Our study also found that long COVID symptoms are considered risk factors for developing mental health symptoms among COVID-19 patients. This finding is consistent with recent studies that have linked long COVID symptoms, such as fatigue, shortness of breath, insomnia, and chest tightness, to a higher risk of depression, anxiety, and stress (44,45). The distress and interference with daily life caused by long COVID symptoms may contribute to the development of mental health problems (46), along with prolonged illness and uncertainty about recovery leading to fear and frustration (47). Additionally, the neurological effects of long COVID may also contribute to mental health issues.]

Risk factors of mental health status and long covid symptoms

[Our study revealed that female COVID-19 patients face a greater risk of developing long COVID and experiencing mental health issues than male patients. This finding aligns with a recent study, which reported that female COVID-19 patients are 3.3 times more likely to experience long COVID than their counterparts (52). The investigation also found that females tend to have a more robust immune response to viral infections, generating higher levels of IgG antibodies than males, which may contribute to a more significant response and increase the risk of long COVID symptoms (53). Furthermore, in Asian societies, women often shoulder a greater burden of domestic responsibilities and caregiving for family members, which can lead to higher levels of stress and psychological distress (54).]

[In our study, we found that individuals with a history of COVID-19 infection are more likely to develop long COVID and mental health symptoms. Such individuals may experience anxiety or depression due to fear of reinfection or the persistence of symptoms (55). Additionally, their weakened immune systems make them more susceptible to long-term health complications, including mental health conditions (47). Patients with a prior history of mental illnesses are also at an increased risk of developing mental health issues, with more than three times the risk compared to those without such a history (56).]

[Moreover, our research indicates that individuals with lower incomes face a higher risk of developing mental health issues and long COVID. Financial stress related to job loss or low income may contribute to the development of mental health issues (57), and limited access to healthcare services could lead to more severe COVID-19 illness and a higher likelihood of experiencing long COVID symptoms (58). Social and economic factors such as housing conditions and accessibility to healthy food and exercise opportunities may also contribute to mental health problems among COVID-19 patients with lower incomes (59).]

_ _ _ _ _ o0o _ _ _ _ _

Reviewer #3: Thank you for providing the opportunity to review this manuscript. The paper presents a potentially exciting study timely on the current problems. This article makes some important contributions in terms of identifying factors associated with long COVID and its association with mental health problems. The results of this study may have important policy implications in its field and may contribute to developing strategies for addressing the issues. However, reading through the paper, I suggest authors improve the manuscript with revisions in some sections to get published.

Many thanks to the reviewer for the comments and suggestions. Please find below a revised manuscript as well as detailed responses to specific comments:

Introduction

The background section is inadequately described. Suggested to expand this section with clear problem statements, rationale and research question/s in a logical flow so that it helps the readers to understand the context and the reasons to conduct this study. Need to describe a bit about mental health situation in Thailand. For example, the information of mental health problem in Thailand before the pandemic. Also, stating the hypothesis or research question/s, and the rationale of the study in this section would be helpful for the readers to understand the study clearly. 

Response - We have now added more text in the introduction as follows:

[Recent research has investigated a correlation between mental health disorders and COVID-19 outcomes, indicating that individuals with pre-existing mood disorders had an increased risk of COVID-19 hospitalization and mortality (4). Throughout the COVID-19 pandemic, there have been conspicuous occurrences of heightened prevalence rates of psychiatric symptoms among the general population worldwide (5). Anxiety, depression, and post-traumatic stress disorder (PTSD) symptoms have emerged as the most prominent and impactful mental health outcomes (6).]

[However, despite many previous studies on the risk factors of mental health status and long COVID among COVID-19 patients, the findings still remain a vague conclusion, warranting further investigation. A recent study found that several certain demographic groups, including women, the elderly, individuals with chronic illnesses, migrant workers, and students, are more vulnerable to developing psychiatric symptoms than the general population (6). Acquiring accurate health information and perceiving the pandemic's impact can contribute to adverse mental health outcomes among COVID-19 patients (10). Conversely, a study conducted in seven Asian middle-income countries revealed that age under 30, a high educational background, being single or separated, and contact with COVID-19 patients are risk factors for mental health issues during the pandemic (11).]

[A recent report conducted in Thailand indicated that the majority of respondents experienced depression, anxiety, and stress (>87.7%) among recovered COVID-19 patients during the pandemic (18). Despite several recent studies on long COVID and mental health disorders, these were primarily review papers that relied on highly heterogeneous studies encompassing different questionnaires, time points, countries, and age groups (19), or were based on hospitalized individuals (20). The results may not accurately represent the experiences of most individuals affected by COVID-19.]

L 92 - The study aims stated is not clear. It differs than previously stated aims in the abstract section. Suggested to make it consistent.

Response – For the objectives of our study, we aimed to: (1) examine the prevalence of long COVID and mental health status among Thai adults who have recovered from COVID-19, (2) identify the association between mental health issues such as depression, anxiety and stress and long COVID symptoms among COVID-19 participants, and (3) investigate the risk factors associated with the correlation between mental health outcomes and the onset of long COVID in adult patients who have previously contracted COVID-19. 

Our objectives stated in the introduction are entirely consistent with the aims in the abstract. In the L92, as mentioned by the reviewer, this is the contribution of our study. In order to avoid misunderstanding for readers, we have now paraphrased the text:

[By shedding light on the impact of mental health on the development of long COVID among Thai individuals, our study offers valuable insights into the contextual information and associated factors concerning mental health status and post-COVID-19 conditions. This contribution is instrumental in developing effective intervention strategies to reduce the risk of long COVID symptoms and address mental health concerns within the Thai population affected by COVID-19.]

Materials and Methods

Reorganization of this section is required. Suggested moving the ethics paragraph at the end of this section. Please note that some Journal asks to move the ethics paragraph under the declaration section – follow the journal structure. However, it is recommended to start the method section with a paragraph describing the study setting/context followed by the study design, sample, and participant recruitment. In the study setting, you may want to provide information about the study area - geographical area, population, the effect of COVID-19 in the area (was the study area a highly affected area?), health facilities and services so on. 

Response – Thanks for the suggestion. According to the Submission Guidelines PLoS ONE journal: “Methods sections of papers on research using human subjects or samples must include ethics statements”. We followed the guidelines. Moreover, before the manuscript was sent to reviewers, the manuscript has passed the editorial check; therefore, we think the structure of the manuscript has fulfilled the requirement.

We have now added more text in the study designed as suggested. Moreover, we have added a figure (Fig 1. Study area in Sichon district, Nakhon Si Thammarat province, Thailand)

[The study was a cross-sectional investigation conducted in November 2022 within the community setting of nine subdistricts in Sichon district of Nakhon Si Thammarat province, southern Thailand (Fig 1). Out of the 10,336 individuals diagnosed with COVID-19 between January 2021 and May 2022 in the databases of a secondary care hospital and three field hospitals, eligible participants included those over 18 years old and had no prior mental health disorder diagnosis by a psychiatrist before contracting COVID-19. Ultimately, a list of 9,396 individuals was considered suitable for participation in our research.]

L 97 - The authors stated that the study protocol adhered to the principles outlined in the Declaration of Helsinki. What are they? Please specify the key principles. 

Response – We followed the guideline for submission to PLoS ONE journal: “All research involving human participants must have been approved by the authors’ Institutional Review Board (IRB) or by equivalent ethics committee(s), and must have been conducted according to the principles expressed in the Declaration of Helsinki” 

Please find the links belows for references: 

(1) Submission Guidelines in PloS ONE journal: 

https://mail.google.com/mail/u/0/#inbox/FMfcgzGsmqxlVxcjltLczlJMNfzNwxNx

(2) Declaration of Helsinki as a statement of ethical principles for medical research involving human subjects: 

https://www.wma.net/policies-post/wma-declaration-of-helsinki-ethical-principles-for-medical-research-involving-human-subjects/

L106 – 107 - This sentence is not related to design. This may go to context/setting. Study design could be Institution or community-based, cross-sectional design…...?

Response – We have added more text in as suggested. 

[The study was a cross-sectional investigation conducted in November 2022 within the community setting of nine subdistricts in Sichon district of Nakhon Si Thammarat province, southern Thailand (Fig 1).]

Sample size and sampling: L109 - 110 stated the approximate range of sample i.e. 900 – 950. What was the sample size calculated? Give the exact number. 

Response – We have now revised the text and provided an exact number of sample size.

[The target sample size was determined using a web-based sample size calculation tool (http://www.winepi.net). The calculation was based on a reported prevalence of 57% of COVID-19 survivors experiencing long COVID (21), a population size (N) of 9,396 from the hospital databases, a margin of error (d) of around 3%, and a confidence interval of 95%. According to the calculation, 942 participants were needed for the study.]

In the result section L 203 stated the sample of 939. How many were approached for interviews and what was the non-response rate? Please describe it clearly. 

Response – After finish data collection and started analyzing data, we found there were 3 cases missing data from respondents. Out of the desired 942 participants, data from 939 individuals were collected. After evaluating the study power of a two-sided significant level 0.05 (p), margin of error 0.03 (d), number of population (N) of 9,396 and sample size (n=939), we found the statistical power (1 – β) was 0.99 (> 0.80). Thus, data from the remaining 939 participants were used for subsequent analyses

We have now revised the text in the manuscript.

[Out of the initial 942 participants, data from 939 individuals were collected and included for further analysis. Missing data was found in three respondents. However, after evaluating the study power, excluding their data maintained a statistical power (99%). As a result, data from the remaining 939 participants were used for subsequent analyses.]

Data Collection: L 165 – 170 - It confuses about the data collection process/procedure. Were the data collection tools and questionnaires self-administered by the participants using online survey form or VHVs collected data conducting face-to-face interviews? If it was self-administered, how BMI was measured? How the measurement accuracy was ensured?

Response – We have now revised the text as follows: 

[The VHVs visited participants’ homes and requested permission to collect data. Each participant was provided a smartphone equipped with an online survey platform to complete the questionnaire. The questionnaire took roughly 20 minutes to complete, and participants could consult with the VHVs if they had any questions during the survey. BMI measurements of participants were obtained using a digital weight scale carried by VHVs during visits to participants' houses to determine their weight. At the same time, the height values mentioned in Thai identity cards were utilized for height measurements.]

Data Analysis: L 192 – stated that the explanatory variables with P < 0.20 in univariate analysis were selected for the final model. Why did you choose the variables with P value < 0.20? Provide the reference for it.

Response – When conducting multivariate analysis, it is often observed that the significance level of several predictive variables may change (i.e., false negative or false positive in the prediction error in regression model due to interactions, confounders). Thus, to ensure that all relevant and potentially predictive variables are examined, we have applied a P-value cut-off in univariable models < 0.2. Any variable with a P-value < 0.2, it will be considered a potential candidate for multivariable models. The multivariable model consisted of variables with P < 0.05 and was used to identify significant risk factors. We have now cited a statistics book and a previous study applied the same method. In the book, please kindly refer to the Chapter 5. 

[Univariable models were initially screened for all explanatory variables, and those with P < 0.20 were selected as candidates for the final model (33,34).]

Results 

The characteristics of study participants and the result descriptions are mixed up that may create confusions to the readers. It is suggested to separate it starting with a paragraph of socio-demographic characteristics of the study participants followed by the result descriptions. 

Response – Thanks for the suggestion, we have now re-written the text with two separate paragraphs as follows: 

[Out of the initial 942 participants, data from 939 individuals were collected and included for further analysis. Missing data was found in three respondents. However, after evaluating the study power, excluding their data maintained a statistical power (99%). As a result, data from the remaining 939 participants were used for subsequent analyses. Most participants were female (77.4%) and younger than 60 (84.7%). Approximately 69.1% of participants were married, and around 80% reported having a high school education or lower. The most common occupation among participants was self-employment (60.7%), and the majority (90%) had a monthly income of less than 15,000 Thai baht (∼450 USD). Regarding BMI, 62.5% of participants were overweight, and 41.9% were classified as obese. Among the recorded historical diseases, hypertension was the most prevalent (18.2%), followed by diabetes (11.7%), dyslipidemia (3.6%), cardiovascular disease (2.8%), allergies (1.5%), and other diseases such as thyroid, psoriasis, asthma, and cancer accounted for less than 1% (Table 1).

Out of the total participants, 745 (79.3%) reported experiencing prolonged COVID-19 symptoms lasting more than two months, while 194 participants (20.7%) did not report any history of long COVID symptoms. Descriptive analyses revealed significant associations between long COVID conditions and gender (χ2 test, p < 0.001), marital status (χ2 test, p = 0.021), monthly income (χ2 test, p = 0.009), BMI (χ2 test, p = 0.014), and any historical diseases (χ2 test, p < 0.001). These findings indicate that these factors significantly contribute to developing long COVID (all p < 0.05) (Table 1).]

Also, a number of repetitions noted in presenting the results. For example, L 203 and L 229 – the same sentence/result is written twice. 

Response – We have now replaced the new sentence: 

[All thirteen symptoms were reported by 745 participants who experienced long COVID conditions. The median number of symptoms reported was 4, with an interquartile range (IQR) of one to seven]

Figure 1 and 2, both are poor and not readable. Could it be presented differently?

A table showing depression, Anxiety and Stress prevalence and severity could be helpful instead of figure. 

Response – For making graphics, our ideas were as follow. 

Figure 1 shows the symptoms characterized as long COVID listed by participants in our study. After examining the data, we found that participants experienced more than one symptom; the median number of symptoms reported was 4, with an interquartile range (IQR) of one to seven. We would not have shown all data if we presented the results as a table or a simple bar chart. We calculated a standardized score for each COVID symptom reported by study participants based on their frequency and percentage of occurrence (as the continuous variables). We showed the data as a box-plot. Since the medians of each symptom were too low, the simple box-plot could not clearly show each symptom of long COVID; we changed a simple box-plot into a violin box-plot to see the different levels of each long COVID symptom.

Figure 2: We completely agree with the suggestion. We have now replaced the Fig 2 by the Table 2 below. 

Table 2. The levels of Depression, Anxiety and Stress among recovered COVID-19 patients.

Levels of Depression, Anxiety, and Stress Number of participant (n = 939) (%)

 Depression Anxiety Stress

Normal 835 (88.9) 760 (84.1) 897 (95.5)

Mild 35 (3.7) 42 (4.4) 25 (2.7)

Moderate 62 (6.6) 111 (11.8) 11 (1.2)

Severe 3 (0.3) 15 (1.6) 5 (0.5)

Extremely severe 4 (0.4) 11 (1.2) 1 (0.1)

Rate from mild to extremely severe 104 (11.1) 179 (19.1) 42 (4.5)

Table 2: L 275. Asterisk (*) indication for p value is not properly done! P<0.01 must have indicated with ** (2 asterisk not 3).

Response – Thanks to reviewer for detecting the mistake. We know have deleted one asterisk at P < 0.01.

Discussion

L 296 – 297 The study has revealed that depression, anxiety, and stress are not prevalent among COVID-19 adults after being discharged from the hospital. This finding or statement is confusing. It doesn’t seem true. Please reconfirm it.

Response – We have now replaced the sentence. 

[Findings from our study revealed a low prevalence of depression, anxiety, and stress among COVID-19 participants.]

_ _ _ _ _ o0o _ _ _ _ _

---

## [Decision Letter · Decision Letter 1]

18 Jul 2023

Prevalence and Factors Associated with Long COVID and Mental Health Disorders among Recovered COVID-19 Patients in Southern Thailand

PONE-D-23-13130R1

Dear Dr. Charuai Suwanbamrung,

We’re pleased to inform you that your manuscript has been judged scientifically suitable for publication and will be formally accepted for publication once it meets all outstanding technical requirements.

Kind regards,

Md. Saiful Islam, BPH, MPH

Academic Editor

PLOS ONE

Additional Editor Comments:

Thanks for addressing all comments and revising the manuscript accordingly.

Reviewers' comments:

Reviewer's Responses to Questions

**Comments to the Author**

1. If the authors have adequately addressed your comments raised in a previous round of review and you feel that this manuscript is now acceptable for publication, you may indicate that here to bypass the “Comments to the Author” section, enter your conflict of interest statement in the “Confidential to Editor” section, and submit your "Accept" recommendation.

Reviewer #1: All comments have been addressed

Reviewer #2: All comments have been addressed

Reviewer #3: All comments have been addressed

2. Is the manuscript technically sound, and do the data support the conclusions?

Reviewer #1: Yes

Reviewer #2: Yes

Reviewer #3: Yes

3. Has the statistical analysis been performed appropriately and rigorously? 

Reviewer #1: Yes

Reviewer #2: Yes

Reviewer #3: Yes

4. Have the authors made all data underlying the findings in their manuscript fully available?

Reviewer #1: Yes

Reviewer #2: Yes

Reviewer #3: Yes

5. Is the manuscript presented in an intelligible fashion and written in standard English?

Reviewer #1: Yes

Reviewer #2: Yes

Reviewer #3: Yes

6. Review Comments to the Author

Reviewer #1: I recommend publication for Predicting outcome with Intranasal Esketamine treatment: a machine-learning, threemonth study in Treatment-Resistant Depression (ESK-LEARNING)

Reviewer #2: (No Response)

Reviewer #3: My comments and concerns are addressed. The manuscript seems much better. I recommend for the publication from my side.

7. PLOS authors have the option to publish the peer review history of their article (what does this mean?). If published, this will include your full peer review and any attached files.

Reviewer #1: No

Reviewer #2: No

Reviewer #3: **Yes: **Hridaya Raj Devkota

---

## [Editor Report · Acceptance letter]

21 Jul 2023

PONE-D-23-13130R1 

Prevalence and Factors associated with Long COVID and Mental Health status among recovered COVID-19 patients in southern Thailand 

Dear Dr. Suwanbamrung:

I'm pleased to inform you that your manuscript has been deemed suitable for publication in PLOS ONE. Congratulations! Your manuscript is now with our production department. 

Kind regards, 

on behalf of

Dr. Md. Saiful Islam 

Academic Editor

PLOS ONE